# Adolescents' screen time displaces multiple sleep pathways and elevates depressive symptoms over twelve months

Sebastian Hökby [1,2*], Jesper Alvarsson [1,2,3], Joakim Westerlund [1,2,4], Vladimir Carli [1,2], Gergö Hadlaczky [1,2]

1 National Centre for Suicide Research and Prevention, Department of Learning, Informatics, Management and Ethics, Karolinska Institutet, Stockholm, Sweden, 2 National Centre for Suicide Research and Prevention, Center for Health Economics, Informatics and Healthcare Research; Stockholm Health Care Services, Stockholm, Sweden, 3 Department of Psychology, Stockholm Centre for Health and Social Change, School of Social Sciences, Södertörn University, Stockholm, Sweden, 4 Department of Psychology, Stockholm University, Stockholm, Sweden

* sebastian.hokby@ki.se

## Abstract

Recently the Swedish Public Health Agency published recommendations of a maximum of two-to-three hours of daily leisure screen time for adolescents aged 13–18, partly to promote better sleep (2024-Sep-02). Biologically and socially, adolescence is characterized by belated sleep times, and depressive effects of screen time can arise through sleep displacements. Theorized links between screen time, sleep, and depression, merited examination of four sleep mediators to determine their relative importance and determine which of them mediate future depression. Hypotheses were preregistered. Three-wave psychometric health data were collected from healthy Swedish students (N = 4810; 51% Boys; ages 12–16; N = 55 schools; n = 20 of 26 Stockholm municipalities). Multiple imputation bias-corrected missing data. Gender-wise Structural Equation Modelling tested four sleep facets as competing mediators (quality, duration, chronotype, social jetlag). The primary model result included the three first mediators to achieve acceptable fit indices (RMSEA = 0.02; SRMR = 0.03; CFI = 0.95; TLI = 0.94). Screen time deteriorated sleep within three months and effect sizes varied between mediators (Beta weights ranged: 0.14–0.30) but less between genders. Among boys, screen time at baseline had a direct adverse effect on depression after twelve months (Beta = 0.02; p <0.038). Among girls, the depressive effect was mediated through sleep quality, duration, and chronotype (57, 38, 45% mediation). Social jetlag remained non-significant. This study supports a modernized 'screen-sleep-displacement theory'. It empirically demonstrates that screen-sleep displacements impact several aspects of sleep simultaneously. Displacements led to elevated depressive symptoms among girls but not boys. Boys may be more prone to externalizing symptoms due to sleep loss. Results could mirror potentially beneficial public health effects of national screen time recommendations.

**Data availability statement:** The study's minimal dataset is located at OSF: Hökby S, Alvarsson-Hjort J, Westerlund J, et al. Screen Time, Depression, and Sleep Associations. DOI: https://doi.org/10.17605/OSF.IO/ZKPVH.

**Funding:** The author(s) received no specific funding for this work.

**Competing interests:** The authors have declared that no competing interests exist.

## Introduction

### Sleep and depression

Depressive and sleep-related disorders during adolescence is a public health concern in many countries [1,2]. In this context, adolescent's use of digital screen technology has also generated interest and concern, partly because both sleep and depressive symptoms seem particularly associated with screen time [2–5]. Previous systematic reviews and meta-analyses concluded that the direction of these associations is unclear [5,6], which has generated further longitudinal studies that shed more light on the causal processes, for example between screen time, sleep [6], depression [7–9], and relevant gender differences [10]. Sleep and depression often coincide [6,11], and both may have consequences for the development of cognitive abilities and emotion regulation abilities [12]. Sleep problems are therefore not only believed to be symptomatic of mood disorders but an underlying cause [11,13]. *Depression* is typically manifested through *somatic-vegetative* expressions such as tiredness, sleep loss, or lack of appetite, in addition to *cognitive-affective* symptoms such as diminished self-image and low affect [10,13,14]. The assessment of these domains is important to clinical scales like to Beck's Depression Inventory-II [14,15], used in this study ("the BDI-II scale").

### Public health recommendations for screen time

Internationally, the Canadian 24-Hour Movement Guidelines for Children and Youth [13] has inspired public health recommendations regarding a <2 *h*/day limit on daily hours of digital media use. The World Health Organization (WHO) 2020 guidelines recognized a <2 *h* screen time limit as favorable to no limit at all, regarding ages 5–17 years [16]. In line with this, countries like Canada [17] and Australia [18] have previously adopted a <2 *h* limit recommendation. However, it has been recognized that the recommendation is far exceeded by about 80% of Australian adolescents aged 16 [18], whereby it becomes questionable whether the recommendation is realistic or idealistic. More recently (2024), Australia proposed a law which would make it illegal for citizens aged <16 to have social media accounts, while France, the United Kingdom, and India, are also considering stricter age-based policies for social media use [19]. Notably, the 2020 WHO guidelines underscored that hourly screen time cut-offs are based on poor quality evidence about the dose-response relationships, and that the safer conclusion to draw is that *lesser is better*. Simultaneously, screen time was pointed out as one of the most problematic of sedentary behaviors, and its potential effects on sleep quality and duration was labeled as an "important" outcome, and depression a "critical" outcome variable [16]. Among Scandinavian countries, Norway has banned smartphone use in middle school classrooms as a public health strategy, with alleged positive heath and academic effects [20]. Danish researchers have examined a <4 *h* limit and found that only 13% of children, aged 6–11, exceeded the limit on weekdays, and 28% on weekend days [21], but have implemented no such strict bans. Neither has this been the case in Denmark's other neighboring country, Sweden, where the intensity of social media use was found comparable to other Scandinavian countries; based on the "2021/2022 Health Behaviour in School-aged Children (HBSC)" survey data, reported by the WHO in 2024 [22].

In September 2024, the Public Health Agency of Sweden (Folkhälsomyndigheten; Report No.: 24161) [23] issued official public health recommendations regarding adolescent digital media use. The recommendations advise the number of hours of daily *leisure* screen time unrelated to school, which should not be exceeded for health reasons. In this context, 'digital media use' (which we translate into 'screen time') refers to social media use, gaming, and streamed multimedia content, but excludes streamed music, podcasts, and other audio. Like the international recommendations [16], the Swedish one's partly rely on parental monitoring,

especially considering parents of children aged 12 or younger. They also explicitly aim to promote better sleep among youth, for example by recommending not to use screens close to bedtime. The recommended cut-off for adolescents aged 13–18 is a <3 *h* limit, and a <2 *h* limit for children aged 6–12 years. The primary focus of this study was the older adolescent group.

The present paper describes a representative sample of 4810 Swedish adolescents, aged 12–16 and studying in Stockholm, where the self-reported leisure screen time on average exceeded the <3 *h*/day recommendation by 1.0 *h*/day. The observed median response was 3–4 *h*/day and its interquartile range (IQR) and standard deviation (SD) can safely be rounded to 1.0 *h* (data collected before Swedish recommendations were issued). Importantly, the purpose of this study was *not* to evaluate the feasibility or quality of these public health recommendations. Instead, it was to examine the extent to which this generally excessive screen time impacted sleep and depression risk in the Stockholm youth population, while recognizing discrepancy between self-reported screen times and new recommendations.

## Natural sleep displacements in adolescence

Healthy adolescent development coincides with a natural sleep behavior change, with a shift towards later sleep times until the age of about twenty [24]. This belated *chronotype* can be expressed, for example, as preferences for later bedtimes, later midpoints of sleep at weekends, and longer sleep duration [25]. This biologically shifted homeostatic sleep pressure is often strengthened through peer relationships, both offline and online. Still, other social conventions, like school start times, tend to remain stable. This gives rise to increased *social jetlag*, when the midpoint of sleep on free days no longer matches the midpoint of sleep required on school days [25]. When less restricted by social obligations in the evenings and at weekends, adolescents are also likely to increase the length of their recreational screen time [26,27]. The most difficult research question, however, is the establishment of causality in screen-related behaviors and subsequent health consequences, and any interacting variables involved [28]. Longer screen use might elevate depressive symptoms either directly, indirectly, or bidirectionally [6,29]. Nevertheless, many published studies have used designs unequipped to test causality or temporality. This study tested effect temporality but not true causality.

## Consequences of sleep displacement

Several studies suggest that screen time might exert its negative mental health effects by displacing healthy sleep behaviors [4,11,30]. We refer here to this phenomenon as 's*creen-sleep-displacement*' – with or without *depressive consequences*. The term 'displacement hypothesis' has been used in explanatory models related to screen time since at least 1988 [31], at a time in history where cut-offs for TV hours were debated much like smartphones are today. In 1989, the well-used Pittsburg Sleep Quality Index (PSQI) was also published [32]. Modern screen time research frequently relies on conceptual replication, extension, and generalizations of the displacement hypothesis, using only some auxiliary hypotheses [33] related to the technology modernizations under scrutiny. Displacement can then happen in several ways; for example, if the screen activity leads to sleep neglect, if the content is stressful, is arousing, or has certain emotional valence, and/or delays the sleep onset through light emissions that breaks down melatonin [34]. As such, sleep displacement and subsequent depression can be viewed as an indirect – *mediated* – consequence of screen time. Displacement effects are supported by studies demonstrating that associations between screen time and depression can be mediated by poorer sleep. The mediating or moderating variable often seems to reflect on socio-demographics, and especially gender, as female gender is more often an effect amplifier in screen time-depression association studies [8,9,35–38]. Previous

studies also suggest that longer screen time can displace sleep and thereby mediate symptoms of inattention and externalizing disorders, which might be more relevant health outcomes for boys. This includes ADHD-associated symptoms, including loss of impulse control and sometimes misconduct behavior [12,36,39,40]. Moreover, excessive screen time may contribute to sedentary lifestyles and unhealthy diets, and harm social relationships, which can confound the associations [29,41]. There seems to be numerous variables which might confound associations between screen use, sleep, and mental health problems. The present study did not attempt to adjust for them, but used a large, well-defined, representative, and randomly selected sample, wherein such confounders can be reasonably assumed to follow a distribution equal to that of the population. However, this assumption was not formally tested for accuracy and might be false, for example with regards to truant behaviors, since our data collections required classroom attendance. Our sample might therefore be slightly healthier than the true average. However, our sampling design is likely to have captured confounders related to socioeconomic differences between Stockholm County municipalities.

## Measuring adolescents' sleep habits

Sleep is a complex and multidimensional habitual human behavior, and there are many ways to measure it. Self-reports are common, and come with corresponding biases (e.g., memory bias). With some positive exceptions (e.g. [34,36,42]), longitudinal studies of adolescent screen time have rarely applied theory-driven Structural Equation Modelling (SEM) to simultaneously examine multiple sleep pathways to depression. More studies have focused on sleep quality, duration, bedtimes, sleep onset delays, and/or daytime tiredness [26]. This selection of sleep parameters has led several studies to conclude that screen use during evenings and night-time is especially problematic for adolescents' sleep [11,34,38]. This notion also converges with our previous cross-sectional findings that sleep quality and duration on school days were the strongest predictors for clinically relevant depression in the current pool of participants [43]. However, it is debatable whether most previous studies have used adequate, valid, comprehensive, and multidimensional sleep measurements, capable of identifying the most screen time- and health-sensitive sleep parameters. Poor measurements would naturally complicate any test aiming to compare health effects of distinct sleep variables. Moreover, considering that most adolescents experience a belated chronotype which makes it natural for them to stay up later in the evening [25,43], the chronotype seems to be an important confounder because it negatively impacts tiredness and sleep drive on school days. Social jetlag is more rarely examined as a competing sleep mediator against sleep times, durations, and quality (relevant exceptions, which also are somewhat incongruent with the official Swedish operationalization of 'digital media use' [23] include [27,41]).

## Aims and hypotheses

This study used a displacement theory-driven multigroup Structural Equation Modelling (gender separated SEM) approach to hypothesis testing. In the context of an association between screen time and subsequent depression levels, four distinct sleep facets were tested simultaneously for their mediating influences on that association [44]. Thirteen distinct sleep parameters (bedtimes on school days, at weekends, and so on) were calculated and consolidated into four sleep domains: *quality*, *average weekly sleep duration*, *chronotype*, and *social jetlag*. Then, five preregistered [45] directional but two-tailed hypotheses (H) were tested for mediation. We expected *positive* longitudinal associations between *longer* screen times (wave 1), *poorer* sleep (wave 2), and *more* depressive symptoms (wave 3). Accordingly, the study assessed the pathways between: (H1) screen time and depressive symptoms; (H2) screen time

and sleep; and (H3) sleep and depressive symptoms. Regarding H1, we further hypothesized (H4) longitudinal, depressive mediation effects of (H4a) *poorer* sleep quality; (H4b) *shorter* sleep duration; (H4c) *later* chronotype; (H4d) *more* social jetlag; and (H5) consistently stronger mediation among *girls* versus boys. The preregistration verifies that hypotheses H4a–H4d are listed from largest to smallest anticipated effect. The study can be said to be a conceptual replication of the old displacement theory [31], applied to modern research findings, adding evidence to the screen health literature. The study is also a 'direct replication' [33] that aims to generalize previously observed findings to the Stockholm adolescent population – partly to test the consistency of the theoretical framework, and partly to assess the degree to which the Swedish screen time recommendations could be relevant to depression health care.

## Materials and methods

### Ethics statement

The study adhered to the Declaration of Helsinki and was approved by the Regional Ethics Committee in Stockholm (Diary number: 2016/2175-31/5). Participants gave informed written consent to participate in research data collection aiming to promote adolescent mental health. Informed written consent was obtained from the parents or legal guardians if the participant was aged <15 years. Participants reporting serious suicidal ideation/planning or attempts within the past two weeks were offered emergency intervention and support.

### Participants and ethical considerations

This prospective observational cohort study analyzed three-wave individual-level data from $N = 4810$ adolescent participants, recruited from $n = 55$ elementary schools in Stockholm County, Sweden. This sample ($N = 4810$) was extracted from a larger pool of student participants ($N = 10\,299$), recruited from $N = 116$ Stockholm schools. Study eligibility and the sample extraction was preregistered (S1 Table). The S1 Table shows that the final sample retained 47% ($n = 4810$ of 10 299) of the original participant pool. Sampling these 55 schools was representative of $n = 20$ of 26 municipalities in Stockholm County. All had a self-reported gender of 'Boy' ($n = 2446$) or 'Girl' ($n = 2364$), and a baseline age of 12–16 years ($M = 14.0$; $SD = 0.72$ years). There was no gender difference regarding baseline age ($t_{(df = 4808)} = 0.087$; $p = 0.931$) nor birth year ($t_{(df = 4808)} = -0.302$; $p = 0.762$). Participants were excluded if they had missing age or gender data or self-reported 'Other Gender' (i.e., not 'Boy' or 'Girl'), or reported a baseline age outside the range of 12–16 years. These exclusion criteria were applied to extract a demographically well-defined sample, recognizing that non-binary individuals are not well-represented. Of note, the study also excluded individuals who reported suicidal plans or attempts within the past two weeks, as all such cases were contacted and offered emergency suicide intervention. Exposure to psychoeducational intervention or suicidal intervention were causes for exclusion as both interventions aimed to alter the mental health trajectory.

### Study design and preregistration

Data were collected between 2016-August-30 and 2019-November-22. The first, fourth and fifth author (SH, VC, GH) had technical access to pseudonymized cross-sectional data throughout the entire collection. The data were primarily collected to evaluate a school-based psychoeducational program in a cluster-randomized controlled trial [46]. This secondary analysis emphasizes a naturalistic design; hence the current sample was characterized by students attending schools randomized to the trial waitlist group. Psychometric scales were administered to participants on digital screen tablets in a classroom setting, supervised by two trained data collectors. Baseline data were collected from students attending 7th or 8th grade.

First follow-up took place after three months, and the last data collection was made 12 months post-baseline, as students transitioned to 8th or 9th grade. The sizable collection of adolescent health data enables several secondary data analyses. The data collection has been registered under the name "*Stockholm school data analysis project*" at the Open Science Framework [47]. The present secondary data analysis study was preregistered as one of its subcomponents: "*Screen Time, Depression, and Sleep Associations*" [45]. The subcomponent stores a comprehensive description of relevant methods and data collections, research questions and hypotheses, data management plans, analytical plan, data processing code, etcetera. The subcomponent also works as a public data repository [45] for storing three cross-sectional datasets and *R*-syntax files, transparently showing our methods for multiple imputation and Structural Equation Modelling (SEM). These datafiles can be used to replicate our Confirmatory Factor Analyses (CFA), conducted prior to hypothesis testing. Furthermore, longitudinal multiple imputation data has been made publicly accessible from the OSF repository (with *mice* dataset identifiers removed). Fig 1 illustrates the hypothesized longitudinal relationships between screen time, sleep habits, and depressive symptoms.

## Depressive symptoms: the outcome

Degree of depressive symptoms was the main outcome variable in this study and were measured using the Swedish translation of Beck's Depression Inventory-Second edition (BDI-II). The scale comprises 21 items which are measured on, or in two cases recoded into, a 0–3 Likert scale [15]. The preregistered internal consistency at wave 3 was acceptable (alpha = 0.91; Total omega = 0.91). When used clinically, the items are summed up to get a total score that ranges 0–63, where the cut-off for 'Mild Depression' is at >13 points; a cut-off we have

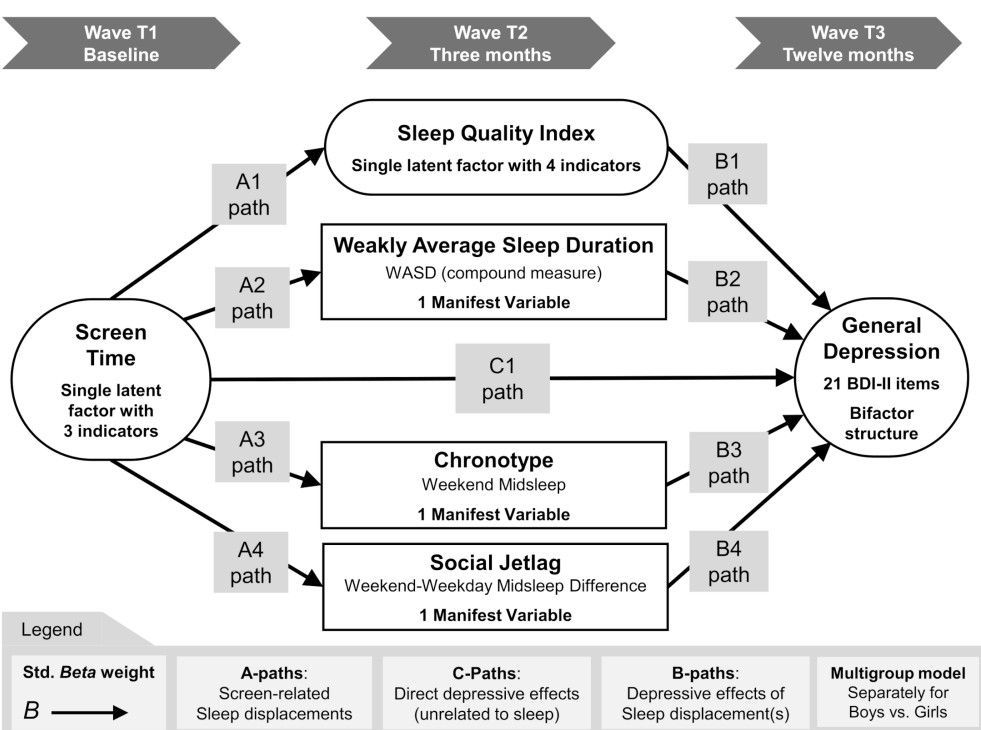

**Fig 1. Simplified Structural Equation Model (SEM).** Indicators and sleep covariances are omitted to avoid clutter.

previously found meaningful to use in the context of sleep and depression in this participant pool [43]. Unfortunately, depression scores must be interpreted differently herein, due to the latent variable modelling. In this study, one could refer to someone's BDI-II score as their "standardized latent G-factor score" (which sounds nonsense to a patient). This is because SEM was used to model the BDI-II scale according to what has been termed the '2+1 bifactor structure of BDI-II' [14]. The bifactor model gets its name from using 10 indicator items to form a Cognitive-Affective (CA) factor, and 11 indicator items to form a Somatic-Vegetative (SV) factor. Together, these two 'bifactors' allow a new General depression factor (+1 G-factor) to be formed from all 21 items at once. This is illustrated in Fig 2 and in the data appendix at the OSF repository [45].

Again, the G-factor was the only outcome of interest, and Fig 2 illustrates this with two grey dashed-dotted arrows that point towards the CA and SV factors but are fixed to zero ($b = 0$). These regressions are *implied* by the model but should *not* be estimated, as they are redundant parameters given the presence of the G-factor. Similarly, the standardization of latent factors was achieved by fixating their variances to one (*Var.* = 1). G-factor loadings (*lavaan* 'std.all' *Beta* weights) were acceptably strong regardless of gender, and overall stronger for boys (*Beta* range: 1.3–2.7) compared to girls (*Beta* range: 1.1–1.8). However, the last indicator item, which loads to the G and SV depression factors, exhibited weaker and almost unacceptable factor loadings (item 21: "Loss of interest in sex"; Boys: *Beta* loading = 0.7; Girls: *Beta* loading = 0.5). This observation replicates two previous analyses of the BDI-II scale conducted on the same data [43,48]. The item was retained to maintain the theory-grounded approach, but it also did not change

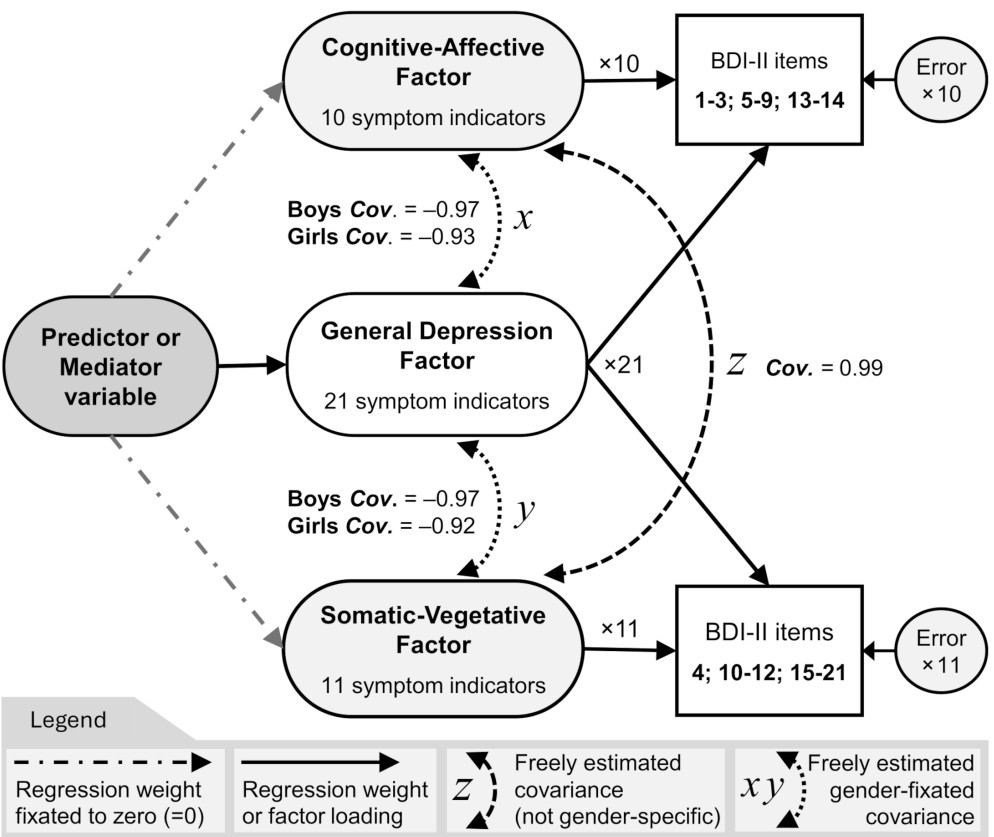

**Fig 2. BDI-II bifactor model.** With estimated factor covariances (*x, y, z*; *p* <0.001).

any conclusions. Lastly, recall that cross-sectional CFA of all relevant scales were conducted as part of the preregistration, to confirm the validity of measurements prior to SEM-based hypothesis testing (visit the project OSF data repository to access the CFA data and *R* code). Important, however, is that the CFA model of BDI-II is a "closed circuit" that cannot be statistically predicted unless some model constraints are modified (such as *Cov.* = 0; *Cov.* = 1; made in CFA), reflected by *R* code modifications. Again, these modifications involved relaxing fixated covariances to instead make free estimations or gender-wise (group-wise) free estimations for the longitudinal SEM. Those modifications, enabling free covariance estimation amongst the BDI-II factors, are illustrated in Fig 2 as three curved, dashed, double-headed arrows ("*Cov. x, y, z*"). They represent covariance estimates which take on large effect values (*Beta* >0.90; *p* <0.001), which Fig 2 displays separately for boys and girls. They are statistically "unwanted" parameters, as they reduce the degrees of freedom in the Chi-square test of local model fit, but they were necessary to include in the present study to achieve acceptable SEM model fit. The primary reason for including covariances between sleep variables is of course because they correlate strongly (or they would not be strictly competing).

## Screen time: the predictor

Previous studies confirm that, as well as screen time, internet use for leisure is also associated with a lack of sleep [49]. Screen time on weekdays and weekend days may also differ [21]. Hence, we used a mixed set of three items that aimed to capture both screen time and internet leisure time. Screen time indicator items were conceived of by the authors. The first item read: "*During a normal day, how much time do you spend watching TV, playing computer games/console games or surfing the internet?*". It was measured on a six-point ordinal scale (response options: <1 *h*, 1–2 *h*, 3–4 *h*, 5–6 *h*, 7–8 *h*, or >8 *h*). The second and third items measured online time on weekdays and at weekends ("*How many hours per day, on average, do you use the internet for leisure activities on Weekdays/Weekends?*"). Responses originally provided on a 0–24-hour integer scale, were later recoded into the ordinal scale of the first item (recoded as: <1 *h;* 1–2 *h*; 3–4 *h*; 5–6 *h*; 7–8 *h*; 9–24 *h*). Ordinal recoding reduced outliers and normalized skewed distributions. In other words, latent screen time was operationalized through indicators of daily leisure internet and screen time per '*School Day*' (Boys = 0.87; Girls = 0.91; *M* score = 3.0), '*Weekends*' (Boys = 0.86; Girls = 0.85; *M* score = 3.7), and '*Normal Day*' (Boys = 0.44; Girls = 0.53; *M* score = 3.0). Schooldays, weekends, and normal days are here ordered from highest-to-lowest factor loading (*Beta* loadings). This way of transforming and combining the three screen time items was performed previously in a similar study [48], and with acceptable psychometric properties. This study, too, noted acceptable internal consistency at the first wave (alpha = 0.77; Total omega = 0.82). The complete case listwise mean ($M_{\text{complete cases}}$ = 3.224) was almost identical to the mean computed from multiple imputation data ($M_{\text{pooled mice}}$ = 3.218). This indicates that the mean and median screen time score was about 3.2 points regardless of gender, corresponding to slightly more than 3–4 leisure screen time hours per day, on a weekly basis. Notably, screen time scores were about the same when estimated from the group of excluded and/or ineligible participants (S6 Table). That group reported similar scores (*M* diff. = 0.09; 95% CI: 0.05 to 0.13; *p* <0.001; Cohen's *d* = 0.09), differing only at the second decimal, implying that the observed screen time estimates reported herein are likely generalizable the entire pool, and not just the waitlist participants (S6 Table).

## Sleep: the mediators

Karolinska Sleep Questionnaire [50–52] was used to measure both sleep *quality* and *quantity*. Firstly, the measure '*Weekly Average Sleep Duration*' (WASD) was derived from the bedtimes,

waketimes, and sleep onset latencies reported for weekdays and weekends separately, and by assuming five weekdays (school/workdays) and two free days per week [21]. Secondly, the *'chronotype'* was defined as the midpoint of sleep at weekends (free days) without accounting for weekly accumulated sleep debt. Thirdly, *'social jetlag'* (SJL) was defined as the difference in midsleep between weekends and schooldays. These calculations were based on previously established definitions of sleep facets, provided by Roenneberg and colleagues [25]. Fourthly, the Karolinska Sleep Questionnaire measured a four-item *Sleep Quality Index* (SQI), which was modelled according to the exploratory and confirmatory factor analysis findings reported by Nordin and colleagues [50], corresponding to the factor loadings reported by Åkerstedt and colleagues [51]. This scale measured the occurrence of disturbed sleep on a 1–6 Likert scale. Higher average scores reflect more frequent symptoms. The recommended screening cut-off is ≥3 points (i.e., at least "3. *Sometimes*: *several times a month*"). Items can be ranked from strongest to weakest factor loadings, using *Beta* values for Boys vs. Girls respectively, as follows: "Repeated awakenings" (0.67 vs. 0.80); "Disturbed sleep" (0.64 vs. 0.77); "Difficulties falling asleep" (0.57 vs. 0.60); "Early awakenings" (0.47 vs. 0.56). The preregistration data appendix reports acceptable psychometrics related to this scale, including acceptable internal consistency (alpha = 0.74; Total omega = 0.74). This SQI variable should not be confused with the Pittsburgh Sleep Quality Index (PSQI) [32]. The subjective ratings on the Karolinska Sleep Questionnaire have shown to correspond decently with polysomnography data in healthy adult sleepers, but age has also been noted to adjust the effect [52]. Thus, the polysomnography correspondence in this adolescent sample is uncertain.

## Statistical analysis

**Software, statistical assumptions, data imputations, and analyses.** Analyses were conducted in *R* (v.4.1.3) through *R* Studio with a conventional significance level (two-tailed alpha = 0.05). The main *R* packages used for data processing and analysis included '*mice* (v.3.16.0)', used for multiple imputation using chained equations [53,54]. Structural Equation Models (SEM) were conducted using '*lavaan* (v.0.6-15)' [55,56] and '*semTools* (v.0.5-6)' [57]. A priori power calculations were made using the add-on 'semTools *SSpower* package'. Psychometrical and descriptive statistics were generated with the '*psych* (v.2.2.9.) package' [58]. The raw data encompassed *n* = 40% (1905 of 4810) longitudinally complete cases. Both occasional non-responses, dropouts and attrition, contributed to data loss. Assuming missingness did not occur completely at random (MCAR = *False*), the *mice* algorithm simulated *m* = 70 sets of imputation data using the Markov-Chain Monte-Carlo method with Predictive Mean Matching (MCMC-PMM). The resulting datasets were used for SEM ($N$ = 4810 × waves = 3 × *m* = 70 sets). This data is available from the data repository with mice identifiers removed for pseudonymization purposes. Models were tested separately for each *mice* dataset using robust maximum likelihood estimation, then using '*semTools*' (lavaan: *MLM*; Satorra-Bentler scaling) to pool models using D2-estimation (test=D2). The resulting output generated several effect estimates, but the most critical ones are the indirect effect and total effect estimate for each mediator, by gender. Dividing any given *indirect effect* (A-path × B-path) by its *total effect* (A-path × B-path + C-path) gives the effect ratio for that mediator. The ratio is interpretable as 'percentage of mediated effects' [59], abbreviated 'PM' below.

**Structural equation model properties.** While primarily using the imputed datasets to achieve maximum power, all final models were validated against the raw complete case data ($n_{complete\ cases}$ = 1905; S2 Table). Interpretations of 'acceptable model fit' were preregistered, and importantly, the preregistered model did *not* achieve acceptable fit, especially not global fit (CFI = 0.46; TLI = 0.38; see S2 Table). Seemingly, this was because *chronotype* and *social jetlag* correlated too highly to co-model (*Beta* >0.85; *p* <0.001). In accordance with the preregistered

protocol for exploratory analyses, a *primary* SEM and an additional *secondary* SEM were run to test all four, not just three, mediators. Both models included SQI and WASD as mediators. The difference is that the primary SEM included chronotype as the third competing mediator, while the secondary SEM tested social jetlag instead of chronotype. No additional multiplicity control was conducted as the preregistration already states the prioritization, i.e., the anticipated effects sizes of sleep variables (SQI> WASD> Chronotype> Social Jetlag), and that we deliberately tested directional hypotheses with two-tailed tests for the sake of estimating with conservative *p*-values. Another important feature is that the entire hypothesis family would become unsupported if just one of the hypothesized associations turn out as significant but in the opposite direction (not equally applicable to the gender difference hypothesis: H5). This "fragility" is an important feature of this theory-driven SEM study, or it would not constitute an appropriate way of testing the feasibility of coherent frameworks.

**Tests of group differences.** We used two procedures to confirm that boys and girls in fact ought to be analyzed in a multigroup setting and not as a single group, considering the probability of encountering gender differences. Firstly, as multigroup SEM was applied to compare boys versus girls, measurement invariance tests were conducted to quantify the degree of gender differences in latent variables (S7 Table). Measurement invariance testing was conducted with semTools (compareFit, test=D2, estimator=ML), after generating CFA models for all latent variables: screen time, sleep quality index, and depression.

For screen time, weak/metric invariance was not found, and partial weak/metric invariance could not be estimated due to the factor only having three indicators, and free estimation of at least two would have been needed. Partial strong/scalar was found with one factor loading and one intercept (*variable:* B53) freely estimated ($F_{(2,3096.3)} = 2.110$; $p = 0.122$).

For the sleep quality index (SQI), partial weak/metric was found with one (*variable:* M47_7) freely estimated factor loading ($F_{(2,541.74)} = 1.446$, $p = 0.236$); partial strong/scalar invariance was found with one loading (*variable:* M47_7) and two intercepts (*variable:* M47_3, M47_7) freely estimated ($F_{(3,547.38)} = 1.389$; $p = 0.245$).

Regarding depression, partial weak/metric invariance was found with five (*Variable:* B, J, M, N, L) factor loadings freely estimated ($F_{(34,80.919)} = 0.203$; $p = 1.0$); partial strong/scalar invariance was found with five loadings (*variable:* B, J, M, N, L) and four intercepts (*variable:* A, D, F, J) freely estimated ($F_{(48,88.901)} = 0.410$, $p = 1.0$).

Taken together these analyses indicated differences in the factor structure of all latent variables between boys and girls. They did not seem to have rated the items in the same way and the assessment of parameters between groups should therefore be done cautiously.

Secondly, gender separation was indicated when comparing the information criteria generated from the multigroup model versus the alternative single group model. Those analyses were based on complete case data, as it generated more information criteria compared to *mice* analysis (S2 and S3 Tables). Since gender separated models consistently yielded better model fit and smaller information criteria than single group models, only gender separated model results are reported and discussed further.

## Results

### Descriptive results

Both genders reported a screen time score corresponding to three-to-four hours of daily leisure screen time. Descriptive statistics are reported in Table 1 below. Average G-factor depression scores were 2.2 times higher for girls than boys (10.1 vs. 4.6 points; CA-factor scores 2.5 times higher; SV-factor scores were 2.0 times higher). Such gender differences in depression have been shown previously [10] and motivate multigroup modelling. Furthermore, girls

reported more SQI symptoms than boys (*Md* symptom score = 2.04 vs. 1.63) and slept for shorter durations. The WASD sleep duration variable indicated that the average complete case participant slept for *M* = 8.22 hours (SD = 1.03 *h*) per night and week. Regardless of gender, participants self-reported a chronotype midsleep at *M* = 5.27 hours past midnight (SD = 1.29 *h*); with a corresponding median social jetlag of 2.33 hours at the end of every week cycle (SD = 1.08 *h*). Compared to girls, boys slept 0.25 hours (15 minutes) longer per night throughout the week (8.35 *h* vs. 8.10 *h*), despite having a 0.29 hour (17 minutes) later chronotype (i.e., a midsleep of 5.42 vs. 5.13 hours after midnight) and despite having 0.24 hours (14 minutes) more social jetlag than girls (2.4 vs. 2.2 hours in weekend midsleep difference).

Exploratory tests of gender differences were conducted for all key variables, and the two depression bifactors. These tests were preregistered without *p*-value correction, as they merely serve descriptive purposes. A series of eight tests were conducted: one independent samples *t*-test (for screen time scores) and seven Mann-Whitney *U*-Tests (for the remaining, more skewed distributions). Gender differences were tested on the complete case data using Listwise deletion. Table 1 shows the sample size for each variable when applying Pairwise deletion. The *t*-test of screen time scores was non-significant (*M* diff. = 0.03; $t_{(df = 1903)}$ = 0.05; *p* = 0.960) but all Mann-Whitney *U*-tests were significant (non-parametric *p* <0.001), implying some degree of gender difference.

## Formal hypothesis testing

Statistically significant *bivariate* C-paths were observed for both genders (Boy's *Beta* = 0.032; *p* <0.001; Girl's *Beta* = 0.054; *p* <0.001). These small effects were generated when the primary SEM was run *without* the four mediators, thus showing that latent screen time scores significantly predicted G-factor depression scores at 12 months. Subsequently, the primary SEM was fitted with the screen time factor as main predictor, the G-factor as main outcome, and with sleep *quality* (SQI), *duration* (WASD) and *chronotype* as competing mediators. Then the secondary SEM model was run by replacing *chronotype* with *social jetlag*. Acceptable compounds

**Table 1. Descriptive statistics. Key variables, including depression bifactors.**

| Measurement | Gender | Pair wise N | Min: Max value | *Skew.* | *Md* | *M* | *SD* |
|---|---|---|---|---|---|---|---|
| Screen Time: Three items, each with a scoring range: 1–6 | Boys | 2154 | 1:6 | 0.52 | 3 | 3.21 | 1.02 |
| | Girls | 2114 | 1:6 | 0.55 | 3 | 3.19 | 0.98 |
| Sleep Quality Index (SQI): Four symptoms scored 1–6 (from best to worst) | Boys | 1496 | 1:6 | 1.50 | 1.5 | 1.63 | 0.64 |
| | Girls | 1584 | 1:6 | 1.27 | 1.75 | 2.04 | 0.91 |
| Weekly Average Sleep Duration (WASD): Hours of duration (*h*) | Boys | 1220 | 3.77: 10.95 | -1.06 | 8.45 | 8.35 | 0.99 |
| | Girls | 1344 | 2.74: 10.46 | -1.10 | 8.24 | 8.10 | 1.07 |
| Chronotype: Weekend Midsleep (Clock time) | Boys | 1286 | 1.88: 10.03 | 0.44 | 5.29 | 5.42 | 1.36 |
| | Girls | 1394 | 1.67: 9.75 | 0.70 | 5.03 | 5.13 | 1.22 |
| Social Jetlag: Number of hourly difference (*h*) | Boys | 1220 | −0.38: 6.96 | 0.58 | 2.33 | 2.46 | 1.12 |
| | Girls | 1344 | −0.67: 6.58 | 0.69 | 2.09 | 2.22 | 1.05 |
| General (G-factor): Depressive symptoms 21 items with total scoring range: 0–63 (Lower is better) | Boys | 1341 | 0:43 | 2.54 | 3 | 4.61 | 6.00 |
| | Girls | 1390 | 0:51 | 1.50 | 7 | 10.05 | 9.44 |
| Cognitive-Affective: Depressive symptoms 10 items with total scoring range: 0–30 (Lower is better) | Boys | 1451 | 0:23 | 2.94 | 0.5 | 1.79 | 3.15 |
| | Girls | 1514 | 0:24 | 1.58 | 3 | 4.45 | 5.04 |
| Somatic-Vegetative: Depressive symptoms 11 items with total scoring range: 0–33 (Lower is better) | Boys | 1452 | 0:22 | 2.08 | 2 | 2.82 | 3.24 |
| | Girls | 1515 | 0:28 | 1.39 | 4 | 5.60 | 4.99 |

of model fit indices were achieved when adding two residual covariances, connecting sleep *duration* (WASD) with *quality* (SQI) and *chronotype* (or *social jetlag* in the secondary model). Fig 3 graphically displays these model modifications as covariances "*Cov.1, Cov.2, Cov.3*", in addition to displaying the main path regression estimates ('Std.all' *Beta* values) generated from the two models (paths A1×B1, A2×B2, A3×B3 and A4×B4 are displayed together in Fig 3). All except the global fit indices exceeded our preregistered cut-off values. Primary SEM model fit: $N = 4810$; $\chi^2 = 1174.2$; $df = 754$; $p < 0.001$; *RMSEA* = 0.010 (RMSEA 90% CI: 0.009 to 0.014); *SRMR* = 0.037; *CFI* = 0.945; *TLI* = 0.936 (preregistered cut-off values for RMSEA/SRMR = 0.08; and CFI/TLI = 0.95). The secondary model had virtually identical fit indices, differing only at the third decimal (S2 Table). Sleep quality, chronotype, and social jetlag, were free to covary with WASD to achieve acceptable fit, indicating that the WASD compound measure of sleep duration was essential to good fit. Recall that WASD incorporates six sleep parameters and is a more informative variable than chronotype and social jetlag. This is to be considered when interpreting the *Cov.1–3* values shown in Fig 3. The WASD-adjustment for sleep onset delays make WASD theoretically distinguishable from the SQI (e.g. problems falling asleep; disturbed sleep).

## Effect interpretations

Detailed statistics and observed effects, including the raw *b*-estimates and their 95% confidence intervals, are detailed in a tabulated format (S4 and S5 Tables). Interpretating Fig 3, notice that screen time predicted sleep measures regardless of gender, except boy's sleep quality (SQI). When using screen time as predictor of sleep at three months, effect sizes related to WASD, chronotype, and social jetlag overlapped both within and between genders (*Beta values raged* = 0.14 to 0.30; $p < 0.05$). As expected, however, the model explained a rather

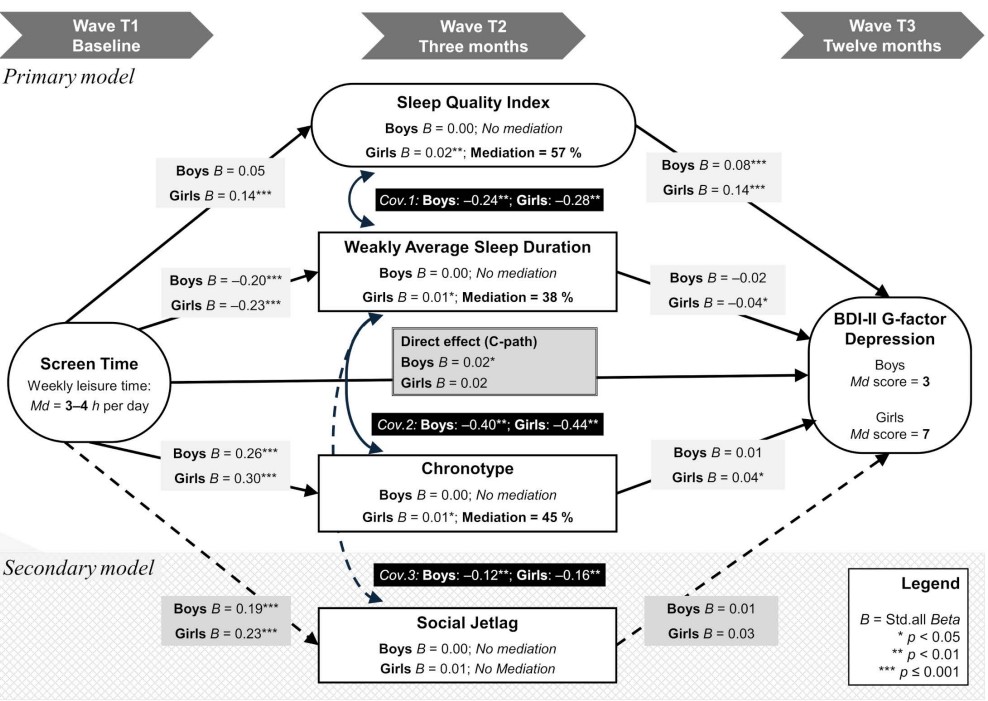

**Fig 3. Main SEM results.** *Beta* effects and mediation percentages.

modest amount of variance in depressive symptoms (Boys: *Beta* = 0.032; *p* <0.001; Girls: *Beta* = 0.054; *p* <0.001). The latent scores are best understood in terms of *Beta* weights and mediation percentages. The effect conversions and ratio calculations are shown separately for boys and girls in S4 and S5 Tables. Moreover, S4 and S5 Tables show that the effect ratio, PM, is calculated from the raw *b*-coefficients (more accurate than using standardized coefficients [59]) and their respective 95% confidence intervals, using three decimals (causing non-substantial rounding errors). Standardized *Beta* values are shown for their ease of interpretation and direct comparison with Fig 3.

As for boys, comparing the second time interval to the first one, results showed that the sleep quality index constituted the only significant B-path (SQI: *Beta* = 0.085; *p* <0.001; WASD: *Beta* = –0.018; *p* = 0.103; Chronotype: *Beta* = 0.014; *p* = 0.128; Social Jetlag: *Beta* = 0.011; *p* = 0.187) yet the only non-significant A-path (SQI: *Beta* = 0.052; *p* = 0.258; WASD: *Beta* = –0.202; *p* <0.001; Chronotype: *Beta* = 0.261; *p* <0.001; Social Jetlag: *Beta* = 0.189; *p* <0.001); leaving no mediation effects at all. S4 Table shows, just like Fig 3, that only a direct effect of screen time on depressive symptoms remained significant among boys (sleep-adjusted *Beta* = 0.021; *p* <0.038).

In contrast to boys, but in line with the gender hypothesis (H5), the screen time effects on girls' depressive symptoms were mediated by all sleep measures except social jetlag (*Beta* = 0.026; *p* = 0.094). No direct effect remained significant (sleep-adjusted *Beta* = 0.015; *p* = 0.323; unadjusted *Beta* = 0.054; *p* <0.001). The C-path effect was relatively small (*Beta* = 0.054), still the longitudinal sleep mediators, measured at three months, explained one-to-two thirds of the total effect. The anticipated order of effect sizes (SQI> WASD> Chronotype> SJL) turned out accurate as well, although the estimated mediation effect for Chronotype was negligibly larger, and virtually equal to WASD (indirect Chronotype *Beta* = 0.011 vs. WASD *Beta* = 0.010; S5 Table). But by far, the SQI exhibited the largest effect size in both the raw/absolute, standardized/relative, and proportional/percentage sense (*b*, *Beta*, PM). The standardized effect, but not effect ratio, was double that of its competing mediators (indirect SQI *Beta* = 0.020 vs. WASD 0.010 vs. Chronotype 0.011, translates to PM = 57% vs. 38% vs. 45% mediation).

As for girls, a comparison of the time intervals shows that the B-path retained an SQI effect virtually identical to its previous effect (*Girls' A-paths*: SQI: *Beta* = 0.141; *p* = 0.001; WASD: *Beta* = –0.234; *p* <0.001; Chronotype: *Beta* = 0.304; *p* <0.001; Social Jetlag: *Beta* = 0.227; *p* <0.001). And while SQI was a stable mediator over time, the B-paths of WASD and Chronotype only retained 18% and 12% of their effects at Wave 3 (*Girls' B-paths*: SQI: *Beta* = 0.142; *p* <0.001; WASD: *Beta* = –0.042; *p* = 0.016; Chronotype: *Beta* = 0.037; *p* = 0.027; Social Jetlag: *Beta* = 0.026; *p* = 0.074).

## Discussion

### Main findings

This naturalistic prospective cohort study tested whether sleep-related behaviors mediated the longitudinal association between screen time and depressive symptoms among 4810 Swedish adolescents, aged 12–16 years. We found support for a preregistered family of directional but two-tailed hypotheses [45]. The hypothesis family stated that these associations, mostly among girls, would be mediated by four deteriorated sleep facets, measured three months post-baseline. The latent screen time score at baseline predicted all facets of sleep, except boy's sleep quality (SQI). These results support studies suggesting that screen time displaces sleep in more than one way, including the duration, quality, and midsleep times.

This conceptually replicates the screen-sleep displacement hypothesis effect, and shows that the hypothesis is theoretically applicable to SEM. It also replicates previous research, and by extension [33], generalize findings directly to the Swedish adolescent population. Among boys, unlike girls, screen-sleep displacement was not associated with depressive symptoms in the subsequent nine months. However, screen-sleep displacement pathways among girls included sleep *quality* (SQI), *duration* (WASD), and *chronotype*. These pathways clearly mediated depressive symptoms in girls, accounting for about half of the association (38–57% mediation). These effects and their temporal direction support previously theorized sleep-mediated relationships between screen time and depression [8,9,30,35–38,41,60]. However, in line with our hypotheses (H5), we only found mediation effects among girls. Our empirical data on boys' depressive trajectories alone, cannot be said to provide direct support for depressive consequences of screen-sleep displacement.

## Validating previous screen-sleep research

Along with a few previous study exceptions [37,41,42], this study validates previous research that has *only* focused on sleep quality and duration, being unable to also measure and analyze additional sleep facets like chronotype and social jetlag (e.g. [61]). Our results suggest that it does not seem too important to always include social jetlag in the typical screen time study. Although the importance of chronotype cannot be neglected due to being the second largest mediator observed in this study, it seems "good enough" to exclude it for many other research purposes, too, if sleep duration (WASD) is fully covered as a weekly cycle. Chronotype could, however, be a possible confounder in some studies that examine night-time screen use [38] and was beneficial but non-essential for good model fit herein. However, more research is needed to show how screen use might impact sleep quality *without* also impacting sleep duration, as this contrast is important to define in future theory development. In this study, poor sleep quality (low SQI score) was characterized by, for instance, repeated awakenings and difficulties falling asleep, and this latent SQI variable seemed to be the most important sleep facet in terms of health effect sizes (*Beta* = 0.02; $p$ <0.01; PM = 57%). This was followed by chronotype (PM = 45%) and weekly average sleep duration (PM = 38%). Our compound measure of sleep duration compressed six sleep parameters into a single manifest variable (WASD = bedtimes, waketimes, and sleep onset delays, for five weekdays and two weekend days, respectively). Because of its inclusiveness of sleep information, WASD became a more essential variable to the SEM when allowing it to covary freely with other sleep facets.

## Gender differences in mental health risks and possibilities

This study notes that longer screen time displaces multiple sleep facets simultaneously. Sleep quality, quantity, and midsleep delays were all affected during the same period. However, the displacement framework we used to guide the SEM procedures claims that sleep displacement should increase symptoms of depression, but never excludes the possibility that it could have mental health consequences other than depression. Although the lack of mediation among boys speaks against the hypotheses (H4a–H4d) of a depressive effect of screen-sleep displacement, it may have other negative health consequences for boys, unmeasured herein. Depression is an internalizing condition that is more prominent among girls [10,62]. Some studies have shown that boys are more prone to develop externalizing problems, such as ADHD symptoms due to screen time [12,36,39,40] and sleep deprivation itself [64].

Internet use offers an "escape", not just in the sense that it facilitates the manifestation of avoidant (escapism-like) coping behaviors [48], but also because internet use narrows the cognitive attention scope, so long as the screen content becomes increasingly fragmented, for

example due to digital multitasking [65]. Thus, it seems possible that self-regulative behavioral problems caused by screen-sleep displacement can have a more immediate depressive effect on girls, while in boys it may first cause impulsiveness, then conduct problems, inattention, or other externalizing symptoms [12,36,39,65], all of which could ultimately lead to depression in the longer term (>12 months). Depression is a more likely outcome if social relationships and other resilience factors suffer because of the screen time. Adolescents' relationships with their teachers, parents, and other adults, could be important health mitigators, but their ability to influence adolescents' screen times are probably limited [9,18], and not sufficient alone [66]. Sometimes parents' strategies to monitor their children's screen time is even hindered by their own screen time desires. This should be met by public health policy which is prepared to postpone school start times [67]. Research shows that 30 minutes of extra sleep in the morning can have substantial effects on alertness and school performance. Our previous cross-sectional findings of sleep and depression associations [43] have further suggested that a 30-minute increase in weekday sleep duration was associated with about 10% lower odds of depression in this sample pool (using the BDI-II cut-off >13). Such structural school interventions and societal shifts are needed if promises of screen time reductions are to be taken seriously. Recall that 30 minutes is twice as much as the gender difference observed in this study (WASD: Boys $M$ = 8.35 $h$; Girls, $M$ = 8.10 $h;$ implies a difference of 15 minutes; Table 1).

Smartphone bans, social media bans, and putting more responsibility on companies to minimize age-inappropriate screen content, might have positive societal effects, but are nevertheless "antagonistic" interventions, based on disagreements. They should not displace interventions which are prosocial or aims to foster digital literacy (author's claim).

## Methodological strengths and limitations

Rather than testing the reversed directionality [59], as we have done previously [48], the present SEM design assumed a certain temporal directionality that turned out to have quite an acceptable model fit. These results strongly support our suggested temporal chain-of-events, despite lacking comparative models. A strength is that the study could incorporate 13 sleep parameters, consolidate them to four main facets, and have them statistically compete almost at once (although a secondary SEM was needed). Confounder adjustments, such as parental monitoring and socioeconomic status and other external influences on screen time and sleep, were not modelled. However, a well-defined sample extraction (regarding gender, birth year, age, grade, and a geographical coverage of 20/24 Stockholm municipalities between which there is socioeconomic variability) likely captures a sizable portion of such error variance. Gender and age are established factors to consider when looking at screen time-related depressive consequences [62]. We therefore think the present findings are likely generalizable. A systematic review [9], and two previous studies of depression in this participant pool [43,48] have indicated *main* but no *interacting* effects of socioeconomic status on depression. Another study has found that objective measures of screen time do not differ substantially depending on socioeconomic factors, compared to gender and age [3]. This could mean that a socio-economic adjustment would not change the pattern of results seen in this study either. This assumption was not formally tested, still we argue that the naturalistic, cluster-randomized study design, likely produced results generalizable to the adolescent population in Stockholm. Undeniably, the data collection required classroom attendance, hence *truancy* – and *non-binary gender* which was excluded from analysis – are cofounder variables which are highly unlikely to have been accounted for by this study.

An independent samples *t*-test indicated that screen time scores in this sample can be internally generalized to the larger pool of participants with valid data (*n* = 9780 of *N* = 10 299; S6

Table). This is a reasonable conclusion as the *t*-test was quite over-powered; that is, the group difference was statistically significant yet negligible in size (Cohen's *d* = 0.09; *p* <0.001). Furthermore, the *mice* algorithm is an appropriate statistical tool for dealing with self-report and self-selection biases, as it robustly accounts for previous responses and neighboring cases when stochastically imputing values [54]. Nevertheless, our sleep and screen-related measurements were still subjective and self-reported, and possibly more biased than "digital phenotyping" procedures that use sleep diaries, actigraphy wristwatches, smartphone trackers, or more advanced technology [3,7,42]. However, it is unclear to what extent this biases the present results.

Considering the generalizable sample of adolescent student participants, the longitudinal design, the preregistered and simultaneous testing of hypotheses using theory-driven SEM, and other methodological procedures this study incorporated to prevent false evidence, we believe that the findings robustly support a general screen-sleep displacement framework. Screen-sleep displacements likely impact sleep habits multifocally, thus explaining relevant developments of depression in youth. Future studies should, however, examine gender separated models with internalizing versus externalizing outcomes and examine how much the screen *content* contributes to the outcome beyond screen *time* [9,36,37,63].

## Implications for screen time recommendations and policies

Sweden's official recommendation of a maximum of two-to-three leisure screen time hours per day [23], is likely to be exceeded by about 1.0 hour (SD = 1.0 *h*) by the average adolescent aged 12–16 years, in the country capital, Stockholm. Although longer than recommended, this average screen time (digital media use) would be even more excessive compared to the WHO recommendation of a <2 *h*/day limit [16], but not compared to the excess screen times observed in other countries of Europe, central Asia, or Canada [22]. The public health recommendation to promote sleep by means of changing screen-related behaviors is both theoretically and empirically supported by this study. It is common for screen time recommendations to acknowledge the importance of sleep in the context of 'screen health'. Screen time, sleep, and depression all have further implications for school performance. Although our study did not examine the appropriateness of the recommended <3 *h*/day limit, our results do suggest that *lesser screen time seems healthier,* in line with previous WHO statements [16]. If screen times were somehow reduced, for example through public health policies [20], our results imply that the high burden of depressive states [1] among young Swedish women, and maybe young men, would likely decrease.

The added value of this longitudinal study of adolescents is that it demonstrated that screen time and internet time can deteriorate sleep within three months, in at least four central aspects simultaneously: sleep quality, duration, chronotype, and social jetlag (Fig 3, paths A1–A4). Considering simultaneously deteriorations of sleep, depression, and screen time, they seem likely to aggravate one another, possibly in a vicious circle. Belated school start times is an evidence-based intervention [67], possibly capable of breaking such spirals. School start shifts may not always be considered a viable public health strategy in the context of screen time. This structural intervention might even seem counter-intuitive since night-time screen time is thought to be especially detrimental to sleep. But the biologically shifted chronotype during adolescence is to be considered a confounder of any 'screen-sleep displacement effects' hypothesized to cancel out the benefits of belated school start times.

## Supporting information

**S1 Table. Sample selection.** Preregistered case extraction procedure.
(PDF)

**S2 Table. Model fit indices.** Preregistered, Primary, and Secondary SEM fit.
(PDF)

**S3 Table. SEM Information Criteria.** Favoring multigroup setting.
(PDF)

**S4 Table. Estimated effect sizes for Boys.** Unstandardized *b*-values and their 95% confidence intervals from which the percentage mediation (PM) is calculated. The standardized *Beta* weights and PM-values are the same as in main manuscript Fig 3.
(PDF)

**S5 Table. Estimated effect sizes for Girls.** Unstandardized *b*-values and their 95% confidence intervals from which the percentage mediation (PM) is calculated. The standardized *Beta* weights and PM-values are the same as in main manuscript Fig 3.
(PDF)

**S6 Table. Screen time estimate bias check.** Scores compared to excluded group.
(PDF)

**S7 Table. Invariance testing.** Detailed results.
(PDF)

## Author contributions

**Conceptualization:** Sebastian Hökby, Jesper Alvarsson, Joakim Westerlund, Gergö Hadlaczky.

**Data curation:** Sebastian Hökby, Jesper Alvarsson, Joakim Westerlund, Vladimir Carli, Gergö Hadlaczky.

**Formal analysis:** Sebastian Hökby, Jesper Alvarsson.

**Funding acquisition:** Vladimir Carli, Gergö Hadlaczky.

**Investigation:** Sebastian Hökby, Vladimir Carli, Gergö Hadlaczky.

**Methodology:** Sebastian Hökby, Jesper Alvarsson, Joakim Westerlund, Gergö Hadlaczky.

**Project administration:** Sebastian Hökby, Vladimir Carli, Gergö Hadlaczky.

**Resources:** Vladimir Carli, Gergö Hadlaczky.

**Software:** Sebastian Hökby, Jesper Alvarsson.

**Supervision:** Jesper Alvarsson, Joakim Westerlund, Vladimir Carli, Gergö Hadlaczky.

**Validation:** Jesper Alvarsson, Joakim Westerlund.

**Visualization:** Sebastian Hökby.

**Writing – original draft:** Sebastian Hökby, Jesper Alvarsson.

**Writing – review & editing:** Jesper Alvarsson, Joakim Westerlund, Vladimir Carli, Gergö Hadlaczky.

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
