## [Decision Letter · Decision Letter 0]

23 Dec 2024

PGPH-D-24-02441

Adolescents’ screen time displaces multiple sleep pathways and elevates depressive symptoms over twelve months

Dear Dr. Hökby,

Thank you for submitting your manuscript to PLOS Global Public Health. After careful consideration, we feel that it has merit but does not fully meet PLOS Global Public Health’s publication criteria as it currently stands. Therefore, we invite you to submit a revised version of the manuscript that addresses the points raised during the review process.

We look forward to receiving your revised manuscript.

Kind regards,

Sok King Ong

Academic Editor

Journal Requirements:

Additional Editor Comments (if provided):

Reviewers' comments:

Reviewer's Responses to Questions

**Comments to the Author**

1. Does this manuscript meet PLOS Global Public Health’s publication criteria ? Is the manuscript technically sound, and do the data support the conclusions? The manuscript must describe methodologically and ethically rigorous research with conclusions that are appropriately drawn based on the data presented.

Reviewer #1: Yes

Reviewer #2: Yes

2. Has the statistical analysis been performed appropriately and rigorously?

Reviewer #1: Yes

Reviewer #2: Yes

3. Have the authors made all data underlying the findings in their manuscript fully available (please refer to the Data Availability Statement at the start of the manuscript PDF file)?

Reviewer #1: Yes

Reviewer #2: Yes

4. Is the manuscript presented in an intelligible fashion and written in standard English?

Reviewer #1: Yes

Reviewer #2: Yes

5. Review Comments to the Author

Reviewer #1: Dear Authors,

I have enjoyed reading the manuscript and I have only a few minor comments and requests for amendments. When my comments or requests for amendments I think this paper can be ready for publication.

Comments om Review of PGPH-D-24-02441

Sentence 145-146: Have they done any analysis to check the distirbution of confounders in the sample vs. the population?

In primary model with the path coefficients you have chosen to present only the p-values for non-statistically significant paths. Please replace the p-values with the path coefficients (but show that it iis not significant by hte lack of a *). so we can see the size of the effect irrespective if it is not statistically significant.

Lines 378-380. Argument is difficult ot follow: How can both two-tailed tests and one-tailed thests inflate type-II error? Compared to each other one of them would inflate the type-II error more than the other? Would not hte difference betweeen one-tailed and two-tailed be that one-tailed used the 95 percentile as a cut-off for «statistical significance» while a two-tailed tests uses the 97.5 percentile as a cut-off? (of course given that the effect is in the expected direction - as you noted in line 382-384.

Line 392-397ff. You write «The results indicated no measurement invariance in screen time, suggesting that the scale would generate a metric value regardless of gender amongst responders». The usage of measurement invariance is used in the opposite way it is meant to. Measurement invariance is found when a scale works the same way for different demographic groups - which is hte opposite way the term is used in the current manuscript. So, in the quited paragraph the scales actually show measurement invariance since the scores are regardless of the respondes gender. It seems to me that the usage of the term invariance varies in the section 392-404. Please ensure that «measurement invariance» is used in the correct way throughout the paper.

In table 1: please indicate the direction of the scales - General (G-factor), Cognitive-Affective and Somatic-Vegetative. Please indicate what is best and what is worst.

line 484 and 489 - and in rest of the paper. Please include the path coefficients also for statistical insignificant paths.

S4Table: Please use same wording for total effects for Social Jetlag as you have for the other time estimates.

S6Table: Please check and correct the 95 % CI for «All cases with valid screen time data» as the 95% CI clearly cannot be [0.05; 0.13] when the mean is 3.12.

Reviewer #2: Dear authors, congratulations for this incredible and well done research.

I do not have much recommendations in the manuscript, otherwise, I suggest to take a look into some grammar mistakes which is common when it comes to not English speakers, and I do definitely understand this.

Besides this, congratulations in this incredible work.

6. PLOS authors have the option to publish the peer review history of their article (what does this mean? ). If published, this will include your full peer review and any attached files.

**Do you want your identity to be public for this peer review?** For information about this choice, including consent withdrawal, please see our Privacy Policy .

Reviewer #1: **Yes: ** Kjell Ivar Øvergård

Reviewer #2: **Yes: ** Jonathan Vicente dos Santos Ferreira

---

## [Decision Letter · Decision Letter 1]

10 Feb 2025

Adolescents’ screen time displaces multiple sleep pathways and elevates depressive symptoms over twelve months

PGPH-D-24-02441R1

Dear Mr Hökby,

We are pleased to inform you that your manuscript 'Adolescents’ screen time displaces multiple sleep pathways and elevates depressive symptoms over twelve months' has been provisionally accepted for publication in PLOS Global Public Health.

Best regards,

Sok King Ong

Academic Editor

Reviewer Comments (if any, and for reference):

Reviewer's Responses to Questions

**Comments to the Author**

1. If the authors have adequately addressed your comments raised in a previous round of review and you feel that this manuscript is now acceptable for publication, you may indicate that here to bypass the “Comments to the Author” section, enter your conflict of interest statement in the “Confidential to Editor” section, and submit your "Accept" recommendation.

Reviewer #1: All comments have been addressed

Reviewer #2: All comments have been addressed

2. Does this manuscript meet PLOS Global Public Health’s publication criteria ? Is the manuscript technically sound, and do the data support the conclusions? The manuscript must describe methodologically and ethically rigorous research with conclusions that are appropriately drawn based on the data presented.

Reviewer #1: Yes

Reviewer #2: Yes

3. Has the statistical analysis been performed appropriately and rigorously?

Reviewer #1: Yes

Reviewer #2: Yes

4. Have the authors made all data underlying the findings in their manuscript fully available (please refer to the Data Availability Statement at the start of the manuscript PDF file)?

Reviewer #1: Yes

Reviewer #2: Yes

5. Is the manuscript presented in an intelligible fashion and written in standard English?

Reviewer #1: Yes

Reviewer #2: Yes

6. Review Comments to the Author

Reviewer #1: I am very happy that you took the time to respond to all my comments and also that you made changes to the manuscript in accordance with my comments. I think the paper can be accepted for publication now.

Reviewer #2: Dear authors

All the reviews and comments were directly accepted and reviewed which means that is all according to the journal.

Best

7. PLOS authors have the option to publish the peer review history of their article (what does this mean? ). If published, this will include your full peer review and any attached files.

**Do you want your identity to be public for this peer review?** For information about this choice, including consent withdrawal, please see our Privacy Policy .

Reviewer #1: **Yes: ** Kjell Ivar Øvergård

Reviewer #2: **Yes: ** Jonathan Vicente dos Santos Ferreira
